# Trends in and Future Research Direction of Antimicrobial Resistance in Global Aquaculture Systems: A Review

**Yayu Xiao [1], Hongxia Wang [2], Chen Wang [3], He Gao [3], Yuyu Wang [1,*] and Jun Xu [2,*]**

1. School of Ecology and Nature Conservation, Beijing Forestry University, Beijing 100083, China
2. Donghu Experimental Station of Lake Ecosystems, State Key Laboratory of Freshwater Ecology and Biotechnology of China, Institute of Hydrobiology, Chinese Academy of Sciences, Wuhan 430072, China
3. Mentougou District Ecological Environment Bureau, Beijing 102300, China; wangdefu1010@163.com (C.W.); 15810957821@163.com (H.G.)
* Correspondence: wangyy@bjfu.edu.cn (Y.W.); xujun@ihb.ac.cn (J.X.)

**Abstract:** The accelerated development of antibiotic resistance genes (ARGs) and antimicrobial resistance (AMR) in aquaculture environments due to the overuse of antibiotics is a global concern. To systematically understand the research trends in and key concepts of ARGs and AMR in aquaculture systems, this study employed the bibliometrix R-package to conduct a bibliometric analysis of the publication characteristics of ARGs and AMR in aquaculture systems from the Web of Science, published from 2000 to 2021. The results revealed that China has produced the most papers. China and the northern hemisphere countries work closely together. Collaboration and multidisciplinary research helped to better understand the impact of AMR in aquaculture on food security and human health. Antibiotic-resistant bacteria and ARGs in aquaculture, as well as the relationship between water environmental variables, antibiotic residuals, and ARGs, are the current research focus. One of the future directions is to establish a conclusive link among water environmental variables, antibiotics, and ARGs. Another future direction is the development of new economical and environmentally friendly technologies to treat AMR in aquaculture wastewater. Collectively, our findings investigate the development directions of AMR research in global aquaculture systems and provide future perspectives.

**Keywords:** bacteria; antibiotic resistance gene; bibliometric analysis; microbial resistance; intestine

## 1. Introduction

Aquatic production plays a vital role in the global food [1,2]. The administration of antibiotics is a common means of controlling diseases in aquatic products in high-density aquaculture environments [3,4]. Aside from therapeutic use, in Africa, many countries have used low concentrations of antibiotics in animal feed to affect their growth performance. Studies have shown that administering chloramphenicol, oxytetracycline, and florfenicol as growth promoters for cultured *Oreochromis niloticus* increased their final weight and growth rate [5–7]. However, recent research has found that this practice has negative effects on nutrient digestibility, digestive enzymes, and the feed conversion ratio [8,9]. Furthermore, studies conducted on zebrafish have shown that oral antibiotics can promote weight gain but damage their immune function [10]. Thus, it is crucial to use antibiotics in animal production judiciously and according to veterinary guidelines. However, antibiotics are often used frequently and in higher doses to ensure their effectiveness is not impacted by exposure methods and time [11]. Due to the unique chemical structures of antibiotics, they are difficult to degrade and decompose in the natural environment and can easily circulate through the food chain before accumulating in the human body [3]. One of the major problems associated with wild antibiotic usage is the rapid increase in the number of antibiotic resistance genes (ARGs) found in the aquaculture environment and intestine of aquatic fauna [2,12–14]. Moreover, antibiotic-resistant pathogens will reduce antibiotic

effectiveness in treating bacterial infections and spread ARGs [13]. Thus, aquaculture systems are increasingly recognized as sources of drug-resistant-bacterial reservoirs of ARGs and antimicrobial resistance (AMR).

Antibiotics play an important role in the rapid emergence of ARGs in aquaculture [15]. When stressed by residual antibiotics, bacteria can pass mobile genetic elements (MGEs), mainly transposons, integrons, plasmids, and insertion sequences [16–18], through horizontal gene transfer to acquire exogenous resistance genes and eventually integrate them into their own genomes to acquire resistance [19–21]. Previous studies have shown that many antibiotics, bacteria, and ARGs in aquaculture provide ideal environments for horizontal gene transfer [2,22]. In addition, antibiotics can accumulate in the environment, and antibiotic resistance genes can be transferred to food webs [23]. The prevalence of ARGs can result in reduced antibiotic efficacy and increased pathogen drug resistance. There is a high risk of food chain transmission, which affects human health. Fifty-two shared ARGs in the intestine of freshwater shrimp and shrimp farms and 32 shared ARGs in the intestine of freshwater shrimp and shrimp farm sediments have been reported [22]. Additionally, some results from a study on zebrafish found that exogenous antibiotic-resistant bacteria (ARBs) could influence the intestinal bacterial community and change its composition, and plasmid-mediated gene transfer can be observed in the intestine of zebrafish [24]. In southeastern Brazil, *Escherichia coli* isolated from surface water suitable for primary contact recreation showed multidrug resistance because of plasmid-mediated gene transfer [25]. Based on this gene transfer mechanism, bacterial strains can acquire resistance to antibiotics even with a small antibiotic content [26]. Therefore, aquaculture systems are predicted to have the potential to store and accumulate AMR, posing a high risk to human health. Aquatic products are recognized as high-protein animal foods, of which 156 million tons of fish were utilized for human food in 2018 [1]. Accordingly, comprehensive studies on AMR in global aquaculture environments are required.

Bibliometric analysis is regarded as an important and effective method for evaluating and assessing qualitative data and quantitative information on research activities [27]. Bibliometric analysis is different from systematic and scoping reviews. It searches databases and uses statistical methods and mapping techniques to perform quantitative and qualitative analyses of specific bibliometric indicators [28]. By extracting information from paper titles, abstracts, affiliations, keywords, collaborative networks, and trend features, quantitative trends are constructed and identified [29,30]. This is an important reference for the layout and analysis of research in related fields. In recent years, scholars have effectively analyzed research hotspots and trends in their respective fields using common bibliometric tools and information, such as authorship, the institution, the country, co-citation analysis, and co-authorship analysis [31,32]. Therefore, we chose this method for our study to analyze big data and future research directions related to AMR aquaculture systems worldwide.

Currently, some studies have been published on various aspects of AMR, such as antibiotic resistance in natural water [33], ARGs in soil [34], ARGs in water removed by metalorganic frameworks [35], AMR from pathogens in the livestock intestine [36], and several others [37]. However, comprehensive reviews focusing on AMR in aquaculture systems are limited. Therefore, this study was based on the global research activity on AMR in aquaculture systems to assess and analyze the following questions: (1) identify the structures in the field of AMR research in aquaculture systems, (2) reveal the development trend in AMR research in aquaculture systems, and (3) discuss future strategic directions for research in this area.

## 2. Materials and Methods

### 2.1. Data Collection

The data in this study were collected from the Web of Science (WOS) database, and a combination of WOS field identifiers and Boolean operators were chosen to form the search formula for collecting the literature. The WOS database was chosen because it is

considered the world's leading database for scientific research assessment, covering the most important and influential research results worldwide [38]. As shown in Figure 1, the search was conducted on 18 January 2022, queried using TS = "antimicrobial resistance gene in aquaculture" OR "antibiotic resistance gene in aquaculture" as the topic, and search terms were "water", "sediment", "organism", and "intestinal microbiota". Then, the timespan was changed to publications from 2000 to 2021, 508 papers were obtained in total, and duplicate papers were deleted. We exported information about the title, author, abstract, and institution and citation information of these papers from the WOS database in bib file format for bibliometric analysis.

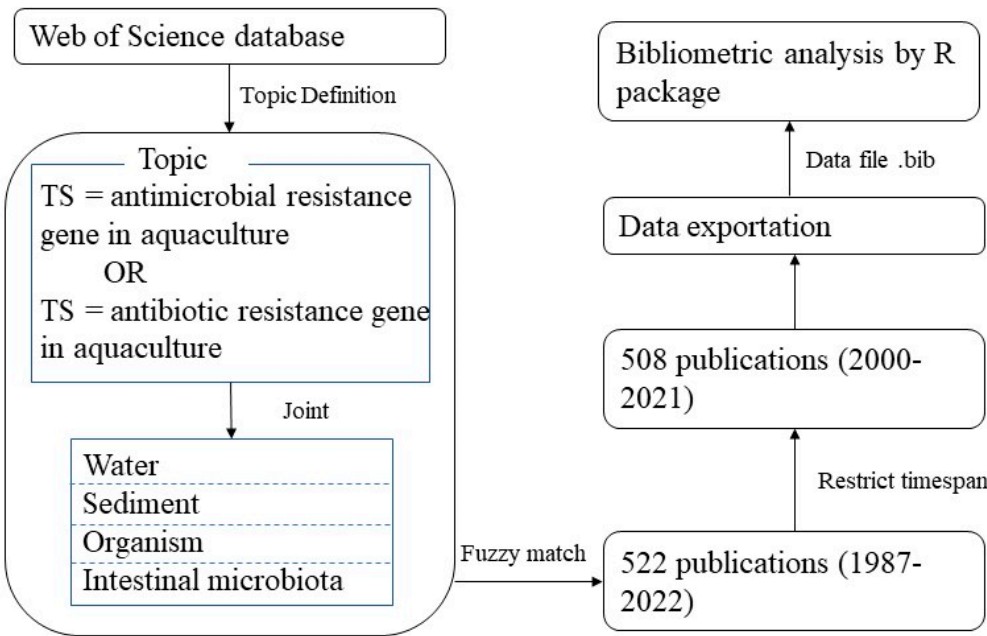

**Figure 1.** Flowchart of aquatic antimicrobial resistance bibliometric analysis.

### 2.2. Methodology

The analysis results of academic documents were visualized using Bibliometrix. Bibliometrix is an important R-tool designed in the *bibliometrix* R-package for comprehensive bibliometric analysis. Bibliometrix could meet the requirements of bibliometric analysis with a process of data import, data transformation, data analysis, and scientific visualization [39]. The target field's literature information was comprehensively and systematically analyzed using citation analysis, graph theory, statistics, network algorithms, factorial analysis, and a thematic map to create various types of knowledge maps that revealed developing research. Bubble diagrams were constructed using the ggplot2 package in R 4.0.3. In addition, 30 recently published papers in 2021 were picked out and used to analyze current research hotspots and future trends.

### 3. Results and Discussion

#### 3.1. Annual Growth of Publications

The search query implemented in the WOS database retrieved the earliest journal articles published in 1997. However, few studies were conducted in the following two years. Therefore, this study focuses on papers from 2000 to 2021. These papers were categorized into four types by analyzing the statistical data. Those retrieved included journal articles (448, 88.12%), reviews (55, 10.83%), conference papers (4, 0.79%), and editorial materials (1, 0.20%).

Figure 2 shows the annual growth of papers on AMR research on global aquaculture systems. The number of publications has increased exponentially over the last 21 years, and as the United Nations' concern about antimicrobial consumption in food animals and

humans began in 2016 [40], more and more scholars have focused on this field. Despite some fluctuations during the period, we found that the cumulative number of papers grew in a quadratic fashion ($R^2$ = 0.9499). And nearly 200 papers were published in every year after 2020. The calculated annual growth rate was 24.3%. This indicates a significant increase in AMR research on aquaculture over the past 21 years, demonstrating the ecological importance of AMR.

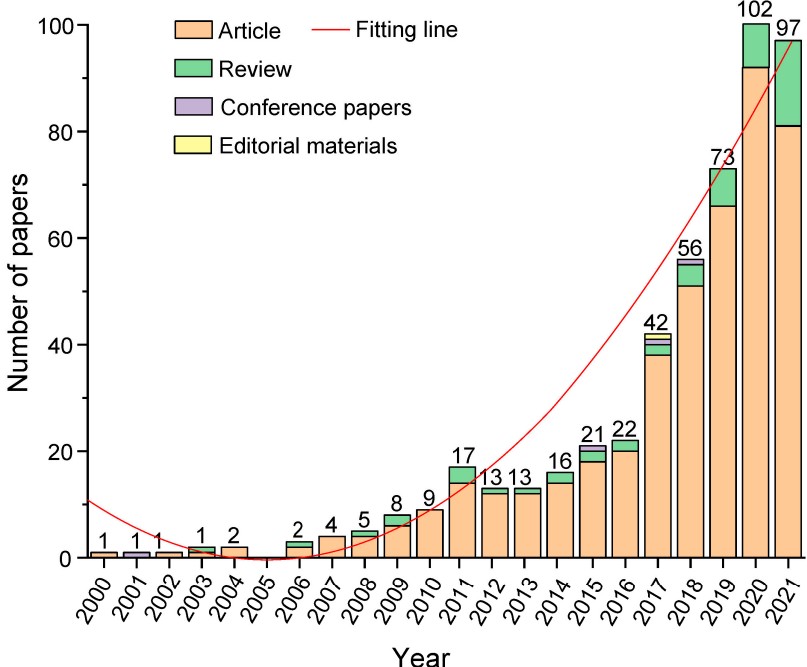

**Figure 2.** Annual growth in number of publications from the Web of Science. The correlation coefficient ($R^2$) of the fitting line was 0.9499.

### 3.2. Regional Analysis of Publications

Scientific cooperation has developed as a result of the growth of contemporary civilization and the expansion of globalization. AMR research on global aquaculture systems was conducted in 61 countries (regions) around the world. Table 1 lists the top 10 countries in terms of output. China had the highest number of published papers (206). The USA and India were ranked as the second and third productive countries, respectively. Interestingly, Spain has only 11 papers in total, and the Average Article Citations values of these articles is extremely high, reaching 114 times, while China and India presented contrary results. A large number of papers could be attributed to antibiotic consumption and fish production in China, which has the world's largest share [1]. Spain tops the average article citations, which is attributable to the paper published by Martinez [41] receiving up 1023 citations.

To better illustrate collaborations between researchers in different countries (regions), co-occurrence networks were constructed (Figure 3). AMR research has been carried out in the northern hemisphere countries (regions) (Figure 3A). The co-occurrence network of countries exhibited 30 nodes and 4 clusters, while the co-occurrence network of institutions showed 19 nodes and 6 clusters (Figure 3B,C). This finding indicates that many researchers from various countries value and collaborate on AMR research on aquaculture systems, which is conducive to the field's rapid development. China had the largest node listed in the largest cluster (red), and it significantly collaborated with most of the countries. Moreover, countries of this cluster were members of the Joint Programming Initiative on Antimicrobial Resistance, which attempts to understand the transmission of AMR and its evolution as priority targets for research [42]. The countries that led in this research field belong to northern hemisphere regions. In addition, 19 institutions in the co-occurrence network are also distributed in the mariculture. Correspondingly, the centralities of nodes

in these countries are greater than zero, indicating that they are critical nodes in the network (Figure 3D). The South China Sea Fisheries Research Institute and the Institute of Cellular and Organismic Biology ranked as the largest in circle size, but their betweenness centralities were low, ranging from 2 to 3. The University of Chinese Academy of Sciences and Institute of Urban Environment were also leading Chinese institutions in this frontier. The University of Copenhagen (Denmark) and University of Helsinki (Finland) were also top research institutions in this field. This finding indicates that, while Chinese scholars published more, the quality of some articles was not exceptional.

**Table 1.** Top 10 countries represented by the papers on AMR research of global aquaculture systems in 2000–2021.

| Country | Papers | Average Article Citations |
|---|---|---|
| China | 206 | 23 |
| India | 35 | 12 |
| USA | 28 | 95 |
| UK | 21 | 82 |
| Republic of Korea | 17 | 19 |
| Japan | 16 | 34 |
| Brazil | 11 | 11 |
| Egypt | 11 | 7 |
| Spain | 11 | 114 |
| Portugal | 10 | 32 |

*3.3. Analysis of Journals and Highly Cited Publications*

The top 10 journals that published the most papers are listed in Table 2. This study used the Hirsch index (h-index), which indicates whether a large number of readers are interested in the subject. Considering the h-index value, the papers published in *Fish & Shellfish Immunology* have had the greatest influence on AMR research in global aquaculture systems from 2003 to 2021, reaching 16. These papers have received 1271 citations. This demonstrates the high impact of papers published in this journal. *Science of the Total Environment* also had an h-index of 16 and it received 1018 citations. Therefore, this finding suggests that AMR research in global aquaculture systems is a research priority. In addition, other fields, such as agriculture, bioengineering, microbiology, and veterinary medicine, appeared in the categories of journals. By combining the advantages of various disciplines and developing various studies and technologies, multiple economic and environmental benefits can finally be realized.

**Table 2.** Top journals based on AMR research of global aquaculture systems in 2000–2021.

| Journal | h-Index | TC | PY-Start | IF2020 |
|---|---|---|---|---|
| Fish & Shellfish Immunology | 16 | 1271 | 2003 | 4.581 |
| Science of the Total Environment | 16 | 1018 | 2011 | 7.963 |
| PLoS ONE | 12 | 760 | 2007 | 3.24 |
| Environmental Pollution | 11 | 1682 | 2009 | 8.071 |
| Aquaculture | 9 | 382 | 2007 | 4.242 |
| Environmental Science and Pollution Research | 9 | 296 | 2015 | 4.223 |
| Frontiers in Microbiology | 9 | 691 | 2012 | 5.64 |
| Chemosphere | 8 | 351 | 2014 | 7.086 |
| Water Research | 8 | 675 | 2012 | 11.236 |
| Ecotoxicology and Environmental Safety | 7 | 218 | 2014 | 6.291 |

Note: TC: total citations; PY-start: published years start; IF2020: impact factor of the journal in 2020.

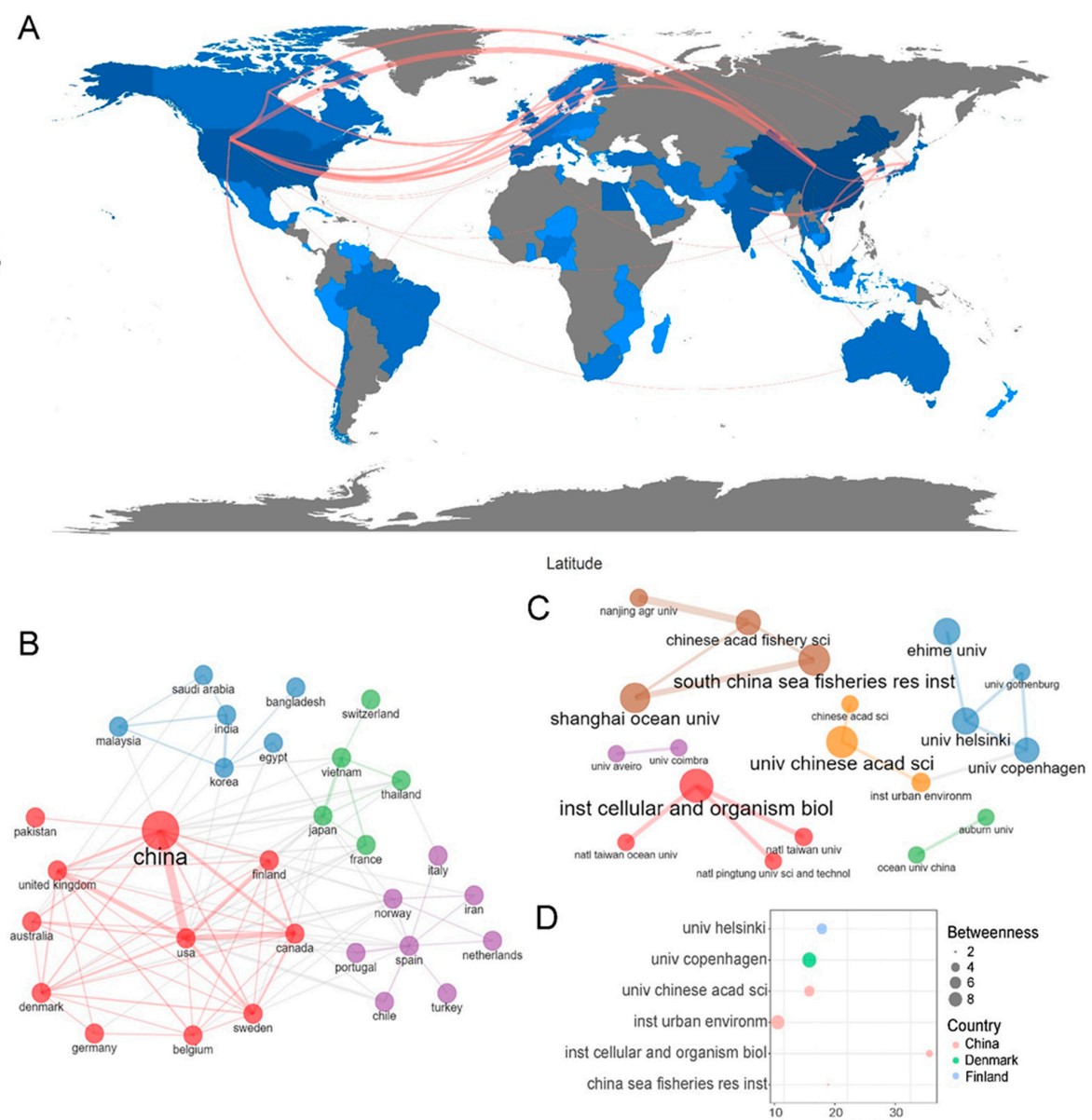

**Figure 3.** Country/region distribution of publications on AMR research of global aquaculture systems in 2000–2021. (**A**) The cooperation map among countries. The red line indicates collaborations between countries. The thicker the line, the greater the collaboration. (**B**,**C**) The cooperation network of countries and institutions, respectively. Continents for the circle color in Figure 2B are listed in Table S1. (**D**) Bubble chart of the publication characteristics for the research institutions (betweenness centrality > 0). The color of the circles represents the continent of each country. The larger the circle, the higher the betweenness centrality of the institution. Abbreviations: inst cellular and organism boil, Institute of Cellular and Organismic Biology; natl pingtung univ sci and technol, National Pingtung University of Science and Technology; natl taiwan ocean univ, National Taiwan Ocean University; natl taiwan univ, National Taiwan University; ehime univ, Ehime University; univ Helsinki, University of Helsinki; univ copenhagen, University of Copenhagen; univ Gothenburg, University of Gothenburg; ocean univ china, Ocean University of China; auburn univ, Auburn University; univ Aveiro, Aveiro University; univ Coimbra, University of Coimbra; univ chinese acad sci, University of Chinese Academy of Sciences; inst urban environm, Institute of urban environment; chinese acad sci, Chinese Academy of Sciences; shanghai ocean univ, Shanghai Ocean University; south china sea fisheries res inst, South China Sea Fisheries Research Institute; chinese acad fishery sci, Chinese Academy of Fishery Sciences; nanjing agr univ, Nanjing Agricultural University.

Citations are used to indicate research hotspots in a certain field. Although citations are not the only way to assess the quality of a paper, they are regarded as an important indicator. A paper can be found, read, and cited by others regardless of whether the opinion about the paper was positive or negative, indicating that the cited paper has reached a level at which it can be referenced or commented on by others. The analysis of the most cited papers should be supplemented by the number of total citations per year to reduce a bias regarding publication time. The top ten cited papers are listed in Table S2. The top ten cited papers were published between 2006 and 2013. These publications mainly focused on AMR research and its influence on human health. Half of these papers are review papers that could effectively summarize previous findings to produce high-quality studies. For example, the top two review papers mainly focused on the abuse and risks of antibiotics in fish aquaculture and the impact of clinical, animal husbandry, and agricultural antibiotic abuse on the antibiotic resistance of microorganisms in the environment.

### 3.4. Development of Antimicrobial Resistance in Global Aquaculture

We performed a timeline view analysis of paper topics from 2000 to 2021, and the trend topics for AMR research on global aquaculture systems are shown in Figure 4. The minimum number of occurrences for each topic was set to 15, leading to the selection of 20 topics for further analysis. However, the number of topics revealed a serious imbalance. Papers in this period were mainly focused on the distribution of and risk of antibiotics to human health [41,43,44]. The research point was then transferred to antibiotic resistance of bacteria based on the development of qualitative PCR. Moreover, "vulnerability" was the longest-running topic from 2012 to 2020. Opportunistic pathogens, such as *Bacillus cereus*, *Bacillus subtilis*, *Bacillus megaterium*, and *Acinetobacter lwofii* [45], were separated. Meanwhile, strategies for manipulating microorganisms or microbial metabolites against pathogenic bacteria were being developed. These measures included third antimicrobial peptides [46] and recombinant proteins [47]. In the following studies, "aquaculture" began to be a hot topic. The fish intestinal immune system was studied [48]. Water, unlike food, can be directly touched by humans. As a result, studies on antibiotics and ARGs in water were constantly evolving. A large number of bacteria carrying ARGs were discovered in water [49–51]. Concerns have been raised about the interactions of new pollutants and antibiotic-resistance genes in water [52]. Furthermore, researchers were interested in the relationship between water environmental variables (chemical oxygen demand, total organic carbon, dissolved organic carbon, suspended solids, and total phosphorus) and ARGs [53], as well as the promotion of bacterial resistance by wastewater discharge from aquaculture systems [54,55].

### 3.5. Current Research Directions of Antimicrobial Resistance in Global Aquaculture

As shown in Figure 5A, factorial analysis divided the keywords into two categories. The red cluster includes traditional antimicrobial resistance research, such as disease, intestinal microbiota, growth, fish, oxytetracycline, sulfonamide, prevalence, etc. Because this area of research is well-established (Table S3), most papers combine these keywords. These studies primarily focused on antibiotics and ARGs in shrimp and fish farm samples (water, sediment [56], feed [57], and intestine samples [58]). One of the antibiotic residual sources was antibiotics in fish feed for disease prevention and treatment in aquaculture [15,59]. Some studies have proved that antibiotics can lead to high ARG abundance [32,60], and the antibiotic concentration has a positive correlation with ARG diversity [49]. Aquaculture activities influence the diversity and abundance of antibiotic resistance genes. While some studies found no correlation between antibiotics and ARG water abundance and diversity, such as in the coastal areas of central Thailand, there was no evidence that ARGs and MGEs in aquaculture were the main drivers of the resistance potential of environmentally resistant microorganisms in central Thailand. In contrast, higher levels of ARGs and MGEs were found in the water at the Hua Krabue canal [61]. Although there is no conclusive evidence linking antibiotics and ARGs, ARGs are widely detected in various

farm or pond regions, such as China, Peru, India, and Chile, etc. [62,63]. As a result, current research has focused on the identification and resistance mechanisms of ARGs. ARGs have been detected in a significant portion of the microbial samples in aquaculture [64]. Many antibiotic-resistant bacteria, for example, were isolated from 12 different shrimp culture ponds in Andhra Pradesh, India [65]. ARGs were found on microbial chromosomes as well [49]. Whole-genome sequencing demonstrated that ARGs confer microbial resistance mechanisms involving antibiotic efflux, antibiotic target alteration, antibiotic inactivation, antibiotic target replacement, and antibiotic target protection [51]. The horizontal gene transfer (HGT) of ARGs can be mediated by mobile genetic elements (MGEs) of microbial communities [16,17,50]. ARGs and MGEs were found to co-occur in marine water from the Gulf of Thailand [62]. However, due to the influence of MGEs, ARGs may not be associated with a specific bacterial class [66]. ARG propagation under selection pressure in aquaculture systems may occur primarily through antibiotic bacterial enrichment and the HGT of ARGs between different bacterial taxa. *Chryseobacterium aquaticum* [50], *Aeromonas hydrophila* [67], *Vagococcus salmoninarum* [68], *Vibrio harveyi*, and *Vibrio parahaemolyticus* [49] were detected ARG fragments. Most of them were cataloged as Gram-negative bacteria [65]. Often these resistant bacteria carry a variety of resistance genes, and 46 ARGs carried by *C. aquaticum*, four ARGs carried by *A. hydrophila* and resistance to erythromycin and eolistin, 13 ARGs carried by *V. salmoninarum*, and nine ARGs carried by *V. harveyi* and *V. parahaemolyticus* were the same but their resistance type of ARG was different. Notably, many antibiotic-resistant bacteria were considered to exhibit genetic relatedness.

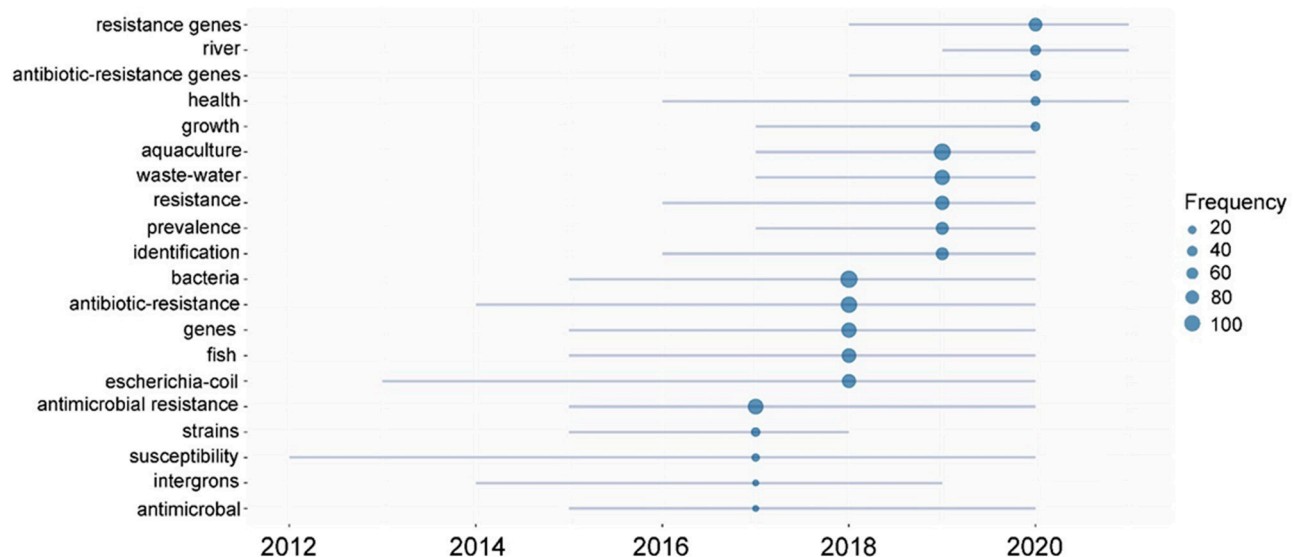

**Figure 4.** Temporal evolution of topics based on AMR research on global aquaculture systems. The *x*-axis represents the timespan, and the *y*-axis displays five keywords that appear with a frequency of at least 15 occurrences per year in the papers.

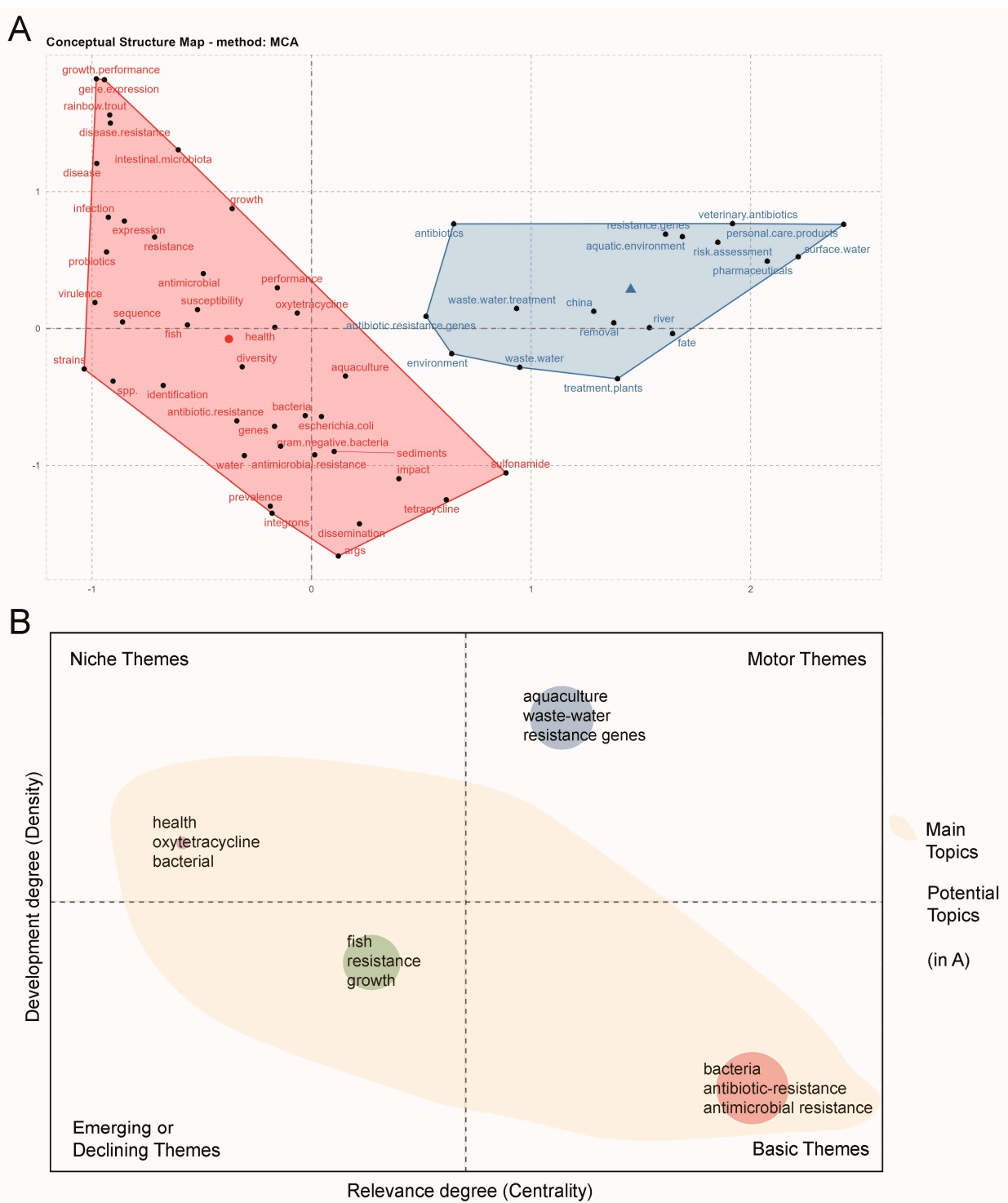

**Figure 5.** Factorial analysis and thematic map of AMR research on global aquaculture systems. (**A**) Factorial analysis plotted based on dimensionality reduction, with a maximum of 50 keywords plus used for plotting. The closer two keywords are, the more papers there are that place them together; the farther they are, the fewer papers there are that place them together. Research on red clusters is mainstream. There are fewer keywords in the blue cluster, indicating that there is still space for further improvement. Moreover, factorial analysis was generated by keywords plus of the WOS database, which is why the word "china" appears. (**B**) Thematic map that converts keyword co-occurrence into centrality and density. The *x*-axis represents the importance of the topic in the research field, as indicated by centrality, while the *y*-axis represents the development of the topic itself, as indicated by density.

*3.6. Futrure Directions*

The keywords in the blue cluster are not strongly linked to traditional antimicrobial resistance research, implying that this section could be improved further (Figure 5A). Moreover, the thematic trend of AMR research in global aquaculture systems is studied using the thematic map, which is determined by the degree of relevance of each keyword within the topic (Figure 5B). Thematic map results are highly consistent with those in Figure 5A. Traditional antimicrobial research topics are located in the first, second, and fourth quadrants. The third quadrant is the emerging or declining themes, represented by the blue cluster in Figure 5. According to the temporal evolution of topics in Section 3.4, it is clear that these topics emerged recently (Figure 4). As a result, these directions will be researched in the future. One of the future directions is to investigate the relationship among antibiotics, ARG, and environmental variables. Many studies have shown that antibiotics [69,70], as well as environmental variables in water, such as chemical oxygen demand (COD) [58], heavy metals (Zn, Pb, Cd, $Cr^{6+}$, and As), and other environmental parameters (permanganate index, pH, and DO), could exert pressure on the spread of ARGs [60]. However, environmental variables affecting ARG are highly variable and lack clear patterns due to the geographical location, culture patterns, timing of collection, and aquaculture organism differences. With more research, the relationship among ARGs, antibiotics, and environmental variables in water might be identified in the future. ARG removal from aquaculture systems is another future direction (Table S3). Antibiotic-resistant bacteria and ARGs [70] can be inactivated by chlorine oxidation [63] and Fe (VI) oxidation [71]. Treatments with ozone nanobubbles can reduce the abundance of *A. hydrophila* [72]. Constructed wetlands with horizontal submerged flow effectively decrease the pathogenic bacteria represented by *Vibrio* [73]. Improved encapsulate fishery drugs based on algal metabolites [74] and the use of probiotics, such as *Phaeobacter inhibens* [75], can all help to reduce the abundance of ARBs and ARGs in the water environment. In addition, *Streptomyces* sp. NHF165 from *Streptomyces* sp. [76], *Bacillus amyloliquefaciens* A23 from *Procambarus clarkii* [77], *Bacillus safenesis* NPUST1 from *Oreochromis niloticus* [78], and mannan oligosaccharide derived from *Saccharomyces cerevisiae* [79] can be used against pathogens of fish intestines as substitutes to antibiotics. The majority of the current experiments are small, expensive, and inefficient. Based on these measures, the applicability of current results to real aquaculture systems and the development of more effective and environmentally friendly removal techniques should be investigated further.

**4. Conclusions**

AMR research in global aquaculture systems is a field of great concern with the number of publications increasing year after year since 2016. The papers from the northern hemisphere mariculture region, particularly in China, have a significant academic influence in the field. AMR research in global aquaculture systems is important for food security and human health, and collaborative research is conducive to achieving multiple economic and environmental benefits. Based on the development of qualitative PCR, the relationship among water environmental variables, antibiotic residuals, and ARGs in aquaculture is constantly evolving, according to the research trend analysis. There is no conclusive evidence linking water environmental variables, antibiotics, and ARGs, which requires further research to resolve. Many antibiotic-resistant bacteria and ARGs were discovered in aquaculture, and removal methods were thoroughly investigated, while these methods are not well applied in practice. Thus, in future research, we should aim to investigate and develop economical and environmentally friendly technologies for the treatment of AMR in aquaculture wastewater.

The analysis in this study is based on the WOS core database, and some Chinese papers were not collected, which may generate bias against other languages. The lack of coverage should be supplemented by searching other databases. The literature in this field provides a good reference for elucidating the evolution of AMR research on aquaculture systems and intestinal ARGs in aquatic organisms worldwide.



**Supplementary Materials:** The following supporting information can be downloaded at: https://www.mdpi.com/article/10.3390/su15119012/s1, Table S1. Continents corresponding to colored circles in Figure 2B. Table S2. Top 10 highly cited papers on AMR research of global aquaculture systems in 2000–2021. Table S3. Thirty papers on AMR research in global aquaculture systems recently published in 2021. References [80–86] are cited in the Supplementary Materials.

**Author Contributions:** Y.X.: Data curation, Methodology, and writing original draft preparation. H.W., C.W. and H.G.: Data curation. Y.W. and J.X.: Conceptualization and writing—review and editing. All authors have read and agreed to the published version of the manuscript.

**Funding:** This study was supported by the National Key R&D Program of China (grant no. 2018YFD0-900904), International Cooperation Project of the Chinese Academy of Sciences (grant no. 152342KYS-B20190025), and National Natural Science Foundations of China (grant no. 31872687).

**Data Availability Statement:** All relevant data are within the manuscript and its Additional files.

**Acknowledgments:** Y.X. thanks all the teachers and classmates in the Ecological Chemometrics group from the Institute of Hydrobiology, Chinese Academy of Sciences, for their support.

**Conflicts of Interest:** The authors declare no conflict of interest.

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
