# Peer review of "Trends in and Future Research Direction of Antimicrobial Resistance in Global Aquaculture Systems: A Review"

_sustainability, doi:10.3390/su15119012_

Round 1
Reviewer 1 Report
In the study entitled “Trends and future areas of research on antimicrobial resistance in global aquaculture systems” the Authors reviewed and summerized the information currently available in the scientific literature on the development of antibiotic resistance genes (ARGs) and antimicrobial resistance (AMR) in aquaculture environments.
The subject of the work is of interest and that the topic of the manuscript falls within the journal topic. Authors rationale is worthy of investigation.
Title is nice and well reflect the rationale of the study, however, Authors should specify the paper is a review, at least in the title.
Lines 80-82: in the sentence “Currently, some studies have been published on various aspects of AMR, such as antibiotic resistance in natural water [26], ARGs in soil [27], ARGs in water removed by metalorganic frameworks [28], and several others [29].” I suggest to add insight on livestock also. Please read and cite the recent paper “Castronovo C. et al., Applied Sciences (Switzerland) 13, (1), 442, 2023.”
Did Authors select exclusion and/or inclusion criteria for the research studies considered?
The figures are nice and well represent the study making it captivating.
Conclusion section is well written. Authors well summarize the main findings on the study’s topic available in the literature and well emphasize the significance of the study.
Reviewer 2 Report
The authors explored the literature on antimicrobial resistance in aquaculture in order to find future research directions using a bibliometric approach. The topic is of interest but, in my opinion, in this paper the subject is treated very lighlty in the introduction. No data are provided on antibiotc use in aquaculture, and by this way the sector could be viewed very negatively. No appropriate data or citations are provided for some very serious statements. For example, at lines 31-33, authors state that many countries supplement animal feed with low concentrations of antibiotics as growth promoters. This is a very serious claim! The use of antibiotics as growth promoters is prohibited in many countries, authors can't affirm this without supporting data.
Even if the analytical approach is new and helps understanding a research topic, the interactions among researchers and possible future trends, this subject should be addressed more carefully. Authors should revise the introduction in order to provide updated and geolocalized information on antibiotic use in aquaculture.
In the present form, the paper is more a methodological exercise.
Reviewer 3 Report
Figure 2. Red line in figure has no legend / explanation.
Table 3 & 4. It is advisable to omit review papers in this study (bibliometric) as the bibliometric manuscript itself is a kind of review paper.
Table 4. Typo southern China, and supposed it should not be called 'situation' but rather 'location'.
3.6 Typography of topic title.
Figure 5 require more explanation on its tendency, centrality & density. No explanation on the figure.
Reviewer 4 Report
The review by Xiao and colleagues used bibliometric analysis of antibiotic resistance publications that had been published over a period. The research is sound and well-presented. But I do have some worries that need to be addressed. Most of these are in the Pdf while some are listed below.
1. All figure legends should contain enough information and be self-explanatory. The authors should take a cue from Figure 3.
2. The authors should consider adding sections to summarize the conclusion of articles in Table 4 based on research directions.
3. Table 4 can be represented by a graph. If the authors want to maintain the table I suggest moving it to the Supplementary file. OR do the following;
a. Title replaced with Type of article, and grouped by review or original article according to Research Direction.
b. The research direction should be followed by a code or a tick at the second and subsequent mention.
4. The authors should edit the article for grammar.
5. Other comments can be found in the Pdf.

Round 2
Reviewer 2 Report
Unfortunately, authors in my opinion underestimated the importance of providing significant and updated literature on the subject of antibiotics use as growth promoters. I suggest, in lines from 34 to 35, to rephrase and specifiy wich countries still lacks a specific regulation on this aspect. With the Regulation (EC) No. 1831/2003 of the European Parliament and of the Council on additives for use in animal nutrition, the EU Commission prohibited the use of antibiotics as growth promoters in animal feeds, for example. Given the delicacy of the subject, is better to be more specific.
Moreover, ref. N. [5] is a quite old citation, cross referencing to other quite old papers (2011 and 2013). Please, search for updated references and provide some data to support your statements. Be more precise on the countries still using this approach to fish growth.
Reviewer 4 Report
The Title “Trends and future areas of research on antimicrobial resistance in global aquaculture systems: A review”
Should be “Trends and future research direction on antimicrobial resistance in global aquaculture systems: A review”
In Figure 2; the explanation for R2 should be given under the figure legend
In Figure 5A, the authors should add a note in the figure legend to clarify the ambiguity for “china”.
Line 330; “Conclusions” should be “Conclusion”
Round 3
Reviewer 2 Report
The manuscript has been improved and can be accepted for publication.